# Water-soluble CoQ10 as A Promising Anti-aging Agent for Neurological Dysfunction in Brain Mitochondria

**DOI:** 10.3390/antiox8030061

**Published:** 2019-03-11

**Authors:** Mayumi Takahashi, Kazuhide Takahashi

**Affiliations:** Biological Process of Aging, Tokyo Metropolitan Institute of Gerontology, Itabashi-ku, Tokyo 173-0015, Japan; ktaka11@tmig.or.jp

**Keywords:** aging, brain mitochondria, water-soluble CoQ10, motor impairment, α-synuclein, oxygen consumption

## Abstract

Mitochondrial function has been closely associated with normal aging and age-related diseases. Age-associated declines in mitochondrial function, such as changes in oxygen consumption rate, cytochrome c oxidase activity of complex IV, and mitochondrial coenzyme Q (CoQ) levels, begin as early as 12 to 15 months of age in male mouse brains. Brain mitochondrial dysfunction is accompanied by increased accumulation of phosphorylated α-synuclein in the motor cortex and impairment of motor activities, which are similar characteristics of Parkinson’s disease. However, these age-associated defects are completely rescued by the administration of exogenous CoQ10 to middle-aged mice via its water solubilization by emulsification in drinking water. Further efforts to develop strategies to enhance the biological availability of CoQ10 to successfully ameliorate age-related brain mitochondrial dysfunction or neurodegenerative disorders may provide a promising anti-aging agent.

## 1. Introduction

Mitochondria in eukaryotes function to regulate cellular metabolism and produce adenosine 5’-triphosphate (ATP) through aerobic respiration [1]. The ATP production is achieved by oxidizing β-nicotinamide-adenine dinucleotide reduced form (NADH) and succinate, which are generated in the mitochondrial matrix as the products of metabolism, followed by sequential electron transfer in the electron transport chain (ETC) through four respiratory enzymatic complexes I-IV. This process of mitochondrial respiration is mediated by two electron transporters coenzyme Q (CoQ) and cytochrome *c*. CoQ-mediated electron transport is catalyzed by two of four complexes embedded in the inner mitochondrial membrane (IMM). In particular, CoQ receives electrons from NADH and succinate via complex I (NADH: ubiquinone reductase) [2,3,4] and via complex II (succinate:ubiquinone reductase) [5]. CoQ then passes electrons to complex III (ubiquinol:cytochrome *c* reductase or cytochrome *bc1* complex) [6,7], which in turn passes the electrons to cytochrome *c*. Complex IV (cytochrome *c* oxidase) [8,9] is the final enzyme in the ETC and receives an electron from each of four cytochrome *c* molecules and transfers the electrons to oxygen molecules, thereby leading to converting into two molecules of water.

Here, we review the recent findings concerning age-associated declines in mitochondrial function and the resultant behavioral and physio-pathological alterations that occur in middle-aged (12–15 months of age) mice. The age-associated declines in mitochondrial function are concomitant with reduced levels of mitochondrial CoQ. Moreover, we show that the age-associated declines in mitochondrial function and CoQ levels are completely rescued by exogenous administration of water-soluble CoQ10. Furthermore, we propose that CoQ is a promising anti-aging agent for neurological dysfunction in brain mitochondria of mice.

## 2. Age-associated Decline in Brain Mitochondrial Function and Motor Impairment

When does mitochondrial function decline and in which organs? Mitochondrial function is maintained by the ETC. As complex IV is the final enzyme in the ETC and transfers the electrons to oxygen molecules, one of the most rapid and reliable methods to evaluate the overall activity of the ETC is determination of the rate of oxygen consumption (OCR) in complex IV. Measurement of the OCR in the liver, kidney, brain, and heart of mice (F1 hybrid of the C57BL/6CrSlc and ICR strains) revealed that complex I and II-mediated OCR are the highest in the heart of both male and female mice [10]. The highest OCR in heart may be reflected by highest mitochondrial DNA content among the tissues and organs, which includes the heart, brain, kidney, liver, red and white muscles, and blood [11]. The only significant difference in the mitochondrial OCR between young (3–4 months old) and middle-aged (12–14 months old) mice was lower complex I and II-mediated OCR in the brain mitochondria of middle-aged male mice than in young male mice [10]. A more precise examination of the age-associated alterations in the brain mitochondrial OCR at three-month intervals using C57BL/6 male mice revealed no significant change in the complex I-mediated OCR until 12 months of age and a significant decline at 15 months of age [12].

The current model for the organization of the mitochondrial respiratory complexes proposes that the different complexes are assembled into supramolecular structures called supercomplexes (SCs) [13,14]. The proposed SCs contain complexes I, III, and IV or complexes III and IV; the association of complex II in SCs has never been described [15]. The respiratory efficiency of mitochondria is regulated by the assembly of SCs [16], and the presence of SCs in the rat heart has been shown to decline with age [17]. Blue native polyacrylamide gel electrophoresis (BN-PAGE) of brain mitochondrial extracts with digitonin [18,19] revealed that complexes I, III, and IV form many types of SCs [14,20]; however, brain mitochondria in middle-aged (15-month-old) male mice with reduced OCR exhibit no age-associated alterations in SC assembly compared to young (six-month-old) mice [21].

Mitochondrial respiratory complexes and SCs localize to the IMM. The dynamin-related protein optic atrophy 1 (OPA1) [22,23] also localizes to the IMM and is essential for IMM fusion [24,25]. In addition to its well-recognized function in regulating mitochondrial fusion, OPA1 participates in remodeling of mitochondrial cristae, SC assembly, and respiratory efficiency [26,27,28]. OPA1 consists of two major isoforms [29,30,31], which have a molecular mass of 95-kDa and 82-kDa and are detected in brain mitochondria of both young (six-month-old) and middle-aged (15-month-old) mice [21]. While the 95-kDa OPA1 is bound to both complexes III and IV, the relative amount of the OPA1 that bound to complex IV, but not complex III, is lower in middle-aged mice than in young mice [21] (Figure 1). By contrast, OPA1 with pathogenic mutations in skin fibroblasts from dominant optic atrophy patients directly interacts with complexes I, II, and III, but not IV [32]. The age-associated reduction in the 95-kDa OPA1 binding to complex IV is accompanied by a significant reduction in the cytochrome *c* oxidase activity of complex IV, however, no remarkable change in the SC assembly is induced [21].

The mitochondrial function is known to decline in several human diseases, including neurodegenerative diseases [33,34,35,36,37,38,39] and neuromuscular disorders [40,41,42]. Parkinson’s disease (PD) is one of the most common neurodegenerative diseases and is characterized by the progressive degeneration of neurons, which is most profound in the substantia nigra pars compacta (SNc) [43,44]. The presence of Lewy bodies (LBs), which are intracytoplasmic proteinaceous inclusions, in surviving neurons of the SNc and other mid-brain regions is a prominent neuropathological feature of PD. LBs consist of fibrillar aggregates of the synaptic protein α-synuclein (α-syn) [43] and most of the protein in LBs is phosphorylated at serine129 (Ser129) [45,46]. Hence, the accumulation of abnormal α-syn, including Ser129-phosphorylated α-syn, in the dopaminergic neurons of the SNc or striatum is considered to cause SNc neurodegeneration in PD patients and PD mouse models [46,47,48,49] through brain mitochondrial dysfunction [50,51,52].

Biochemical analyses indicate that the levels of Ser129-phosphorylated α-syn are significantly higher in the brains of middle-aged mice than in the brains of young mice. Histological analysis of brain sections revealed a significant increase in the number of Ser129-phosphorylated α-syn-positive neurons in the motor cortex, but not in the nigrostriatum of middle-aged mice. Moreover, the increased Ser129-phosphorylated α-syn accumulation in the motor cortex was accompanied by a significant reduction in the levels of vesicular glutamate transporter 1 (VGluT1), a maker of glutamatergic neurons [53,54,55,56], in the motor cortices.

Mitochondrial dysfunction in PD is frequently accompanied by motor impairment, which is characterized by the cardinal motor symptoms of resting tremor, rigidity, and bradykinesia, and refers to a slowness of movement [57,58,59]. Several behavioral tests, including the pole test (PT) [60,61,62,63], open-field test (OFT) [64,65,66], and home-cage test (HCT) [65,67,68], indicate that middle-aged (15-month-old) mice exhibit a slowness of movement and reduced exploratory and voluntary locomotor activities compared to young (six-month-old) mice [12]. However, general exploratory activity is not significantly different between young and middle-aged mice [12].

α-Syn usually localizes to synaptic vesicles (neurotransmitter vesicles) and plays a role in neurotransmitter release from the synaptic membranes via modulating vesicle pool and vesicle docking with the membrane. On the other hand, VGluT1 is preferentially associated with the membranes of synaptic vesicles and function in glutamate transport [53,54,55,56]. Accumulation of abnormal α-syn in PD interferes with the priming of synaptic vesicles, which leads to a decrease in the size of the pool of releasable vesicles [69]. Moreover, the motor cortex is upstream of the corticostriatal circuit or cortex-nigrostriatal circuit connection [70,71,72]. Taken together, brain mitochondrial dysfunction that is accompanied by increased phosphorylated α-syn levels together with decreased VGluT1 levels in the motor cortex is suggested to result in motor impairment of middle-aged mice.

## 3. Water-soluble CoQ10 Rescues Brain Mitochondrial Dysfunction and Motor Impairment

What is the reason for the reduced mitochondrial function in middle-aged mice? An age-associated decline in brain mitochondrial function is thought to be due to defects in the ETC. In the brains and platelets of PD patients, a significant decrease in mitochondrial respiratory complex I activity is accompanied by reduced levels of CoQ [40,41,42,73,74]. Biosynthesis is the main source of CoQ, which is catalyzed by at least 10 enzyme proteins encoded by *coq* genes (*coq*) [42]. The *coq-7/clk-1* gene, which has been identified as a particularly critical *coq* gene, encodes a demethoxyubiquinone mono-hydroxylase that functions in the penultimate step of CoQ biosynthesis [75,76]. Therefore, *coq7/clk-1*-deficient animals lack CoQ [77,78,79], and this deficiency is embryonic lethal without exogenous CoQ10 supplementation [78,79,80].

Apart from decreased CoQ levels due to loss or mutation of the *coq* genes, the levels typically decline during the aging process [81,82,83]. The primary types of CoQ in mice are CoQ9 and CoQ10 and the ratio of CoQ9 to CoQ10 is 2.4 in young mice and 2.5 in middle-aged mice [12]. The results indicate no significant change in the ratio in mouse brains with age and agree with the results from rat [83]. The oxidized forms of which (Q_9_ and Q_10_) use complexes I and II to transport electrons to complex III and become the reduced forms of CoQ9 (Q_9_H_2_) and CoQ10 (Q_10_H_2_) [2,3,5]. The amounts of Q_9_H_2_, Q_9_, and total CoQ9 (Q_9_H_2_ plus Q_9_) in brain mitochondria of middle-aged (15-month-old) mice are significantly lower than those in young (six-month-old) mice [12]. Similarly, the amounts of Q_10_H_2_, Q_10_, and total CoQ10 (Q_10_H_2_ plus Q_10_) in middle-aged mice are significantly lower than those in young mice. Contrary to these, the ratios of Q_9_H_2_ to Q_9_ and Q_10_H_2_ to Q_10_ are comparable between young and middle-aged mice [12]. These results indicate that the levels of both the reduced and oxidized forms of CoQ9 and CoQ10 and total CoQ9 and CoQ10 are significantly lower in the brain mitochondria of middle-aged mice than in the brain mitochondria of young mice.

It is well known that CoQ is highly hydrophobic, so that its water solubility is critical for restoration of mitochondrial function by CoQ10 supplementation. Several formulations of water-soluble CoQ10 ameliorate behavioral defects in animal models of PD [84,85] or renal disease in a mouse model [86] as an antioxidant. However, there are very few experimental data about the effect of water-soluble CoQ10 on mitochondrial function in mice or rat [87], except for cultured cells [88,89]. Among many water-soluble CoQ10 formulations, CoQ10 nanoemulsion with a mean diameter of 52 nm dissolved in glycerol-fatty acid-based solvent could rescue CoQ-deficient cell survival via restoration of mitochondrial function [80,90]. The administration of this formulation of water-soluble CoQ10 to middle-aged (15-month-old) mice via drinking water significantly increased the amounts of Q_10_H_2_, Q_10_, and total CoQ10 to levels comparable to those in brain mitochondria of young mice [12]. Similarly, the amounts of Q_9_H_2_, Q_9_, and total CoQ9 in the middle-aged mice are significantly increased by water-soluble CoQ10 administration to the levels comparable to those in young mice. In contrast, exogenous CoQ10 administration to young (six-month-old) mice significantly reduces the amounts of both Q_9_H_2_ and Q_10_H_2_ in the brain mitochondria; however, the amounts of Q_9_ and total CoQ9 and Q_10_ and total CoQ10 remained unchanged. These results indicate that exogenous water-soluble CoQ10 successfully passes through the blood-brain barrier and enters into brain mitochondria, thereby leading to elevating mitochondrial levels of CoQ9 and CoQ10 in middle-aged mice, but not in young mice. Approximately 0.3%~0.5% of exogenous CoQ10 is incorporated into brain mitochondria after the administration of 150 μM water-soluble CoQ10 via drinking water for seven days [12].

While the administration of exogenous water-soluble CoQ10 to young mice does not affect the complex I or complex II-mediated OCR in brain mitochondria, CoQ10 administration to middle-aged mice via drinking water significantly increases the complex I-mediated OCR to levels more than 3-fold greater than those in control middle-aged mice [12]. Similarly, the complex II-mediated OCR significantly increases with CoQ10 treatment in middle-aged mice and to levels more than two-fold higher than that in the control mice. Therefore, water-soluble CoQ10 administration not only rescues the reduced complex I-mediated OCR, but also enhances the complex II-mediated OCR in middle-aged mice.

Regarding the effects of exogenous CoQ10 on OPA1, the administration of exogenous water-soluble CoQ10 to middle-aged mice via drinking water induces no remarkable change in total OPA1 (95-kDa and 82-kDa OPA1) or 95-kDa OPA1 levels in brain mitochondria. However, the relative amount of 95-kDa OPA1 bound to complex IV is significantly higher in middle-aged mice given CoQ10 via drinking water than in control middle-aged mice. Moreover, the reduced cytochrome *c* activity of complex IV in middle-aged mice is significantly enhanced by exogenous water-soluble CoQ10. Taken together, these observations indicate that water-soluble CoQ10 rescues the aged-associated declines in OPA1 binding to complex IV and the cytochrome *c* oxidase activity of complex IV (Figure 1).

OPA1 is a member of the mitochondrial dynamin family of GTPases [22,23], including OPA1, dynamin-related protein 1 (DRP1), mitofusin 1 (MFN1), and MFN2 [91]. Among these, OPA1 and DRP1 have a GTPase effector domain (GED) [92,93]. As the CoQ10-induced restoration of brain mitochondrial OCR in middle-aged mice is completely inhibited by 15-deoxy-prostaglandin J_2_ (15d-PGJ_2_), an inhibitor of GED-containing GTPases [93], but not Mdivi-1, a selective inhibitor of DRP1 [94,95], the CoQ-responsive restoration of OCR requires the GED-containing GTPases activity of OPA1. Similarly, the exogenous CoQ10-induced enhancement of OPA1 binding to complex IV and restoration of complex IV cytochrome *c* oxidase activity in brain mitochondria of middle-aged mice are inhibited by 15d-PGJ_2_. The results indicate that the GTPase activity of OPA1 is crucial for OPA1 binding to complex IV and restoration of complex IV activity, thereby restoring the mitochondrial OCR in middle-aged mouse brain [21].

Is restoration of mitochondrial function, which includes mitochondrial CoQ levels, OPA1 binding to complex IV, and cytochrome *c* oxidase and oxygen consumption activities of complex IV, by water-soluble CoQ10 supplementation accompanied by amelioration of neurological alterations in the motor cortices? Administration of water-soluble CoQ10 via drinking water significantly decreases the number of Ser129-phosphorylated α-syn-positive neurons and increases the VGluT1 levels in the motor cortices of middle-aged mice to levels comparable to those in young mice [12]. These results indicate that neurological defects in the motor cortices of middle-aged mice are completely rescued by exogenous water-soluble CoQ10 in drinking water. Similarly, dietary administration of CoQ10 exhibits neuro-protective effects against induction of α-syn aggregates and loss of dopamine in a mouse model of PD [96].

Are the rescues of increased accumulation of Ser129-phosphorylated α-syn and decreased VGluT1 levels in the motor cortex in middle-aged mouse brains by exogenous administration of water-soluble CoQ10 accompanied by a restoration of motor function? Indeed, water-soluble CoQ10 in the drinking water of middle-aged (15-month-old) mice completely ameliorated the reduced motor activities, including the rate of movement and voluntary loco-motor activities, but had no effect on general exploratory activity [12]. In animal models of PD, behavioral impairment and pathological defects are ameliorated by water-soluble CoQ10 as an antioxidant [84,85].

## 4. Conclusions

Mitochondrial function declines during the normal aging process [10,12] and in several neurodegenerative [33,34,35,36,37,38,39] and neuromuscular diseases [40,41]. Mitochondrial dysfunction includes declines in the expression and activity of respiratory complexes, oxygen consumption, and ATP production and is caused by defects in the ETC, including electron transporters, such as CoQ. The age-associated mitochondrial dysfunction and the accompanying behavioral and pathophysiological defects due to the CoQ deficiency are rescued by oral administration of water-soluble CoQ10, as described above [10,12,21].

Many reports have examined the beneficial effects of CoQ10 in aging and age-related disorders as an antioxidant [84,85,86,88,89,97,98]. However, improvement of the bioenergetics process of mitochondrial dysfunction via oral administration of CoQ10 requires overwhelming many difficulties, such as CoQ10 passing through the blood-brain barrier and the cytoplasmic and mitochondrial membranes of target cells after its absorption from the small intestinal tract [99,100]. To overcome these difficulties, water solubility and the size of CoQ10 nano-emulsion are considered critical for determining the rate and extent of drug release [99,100,101,102]. Recent studies have developed various formulations of CoQ10 with improved solubility and biological availability that exhibit excellent antioxidant activity [86,97]. Furthermore, the water-soluble CoQ10 nanoemulsion is incorporated into brain mitochondria and rescues brain mitochondrial dysfunction and behavioral disorders in normal aging mice [12] and in animal models of PD [84,85] or encephalopathy [87]. There are a few therapeutic data about the effects of water-soluble CoQ10 on human disorders [103,104,105]. These reports indicate that water-soluble CoQ10 is a promising anti-aging agent that has the potential to restore mitochondrial function for the treatment of age-related mitochondrial dysfunction and neurodegenerative diseases.

## Figures and Tables

**Figure 1 antioxidants-08-00061-f001:**
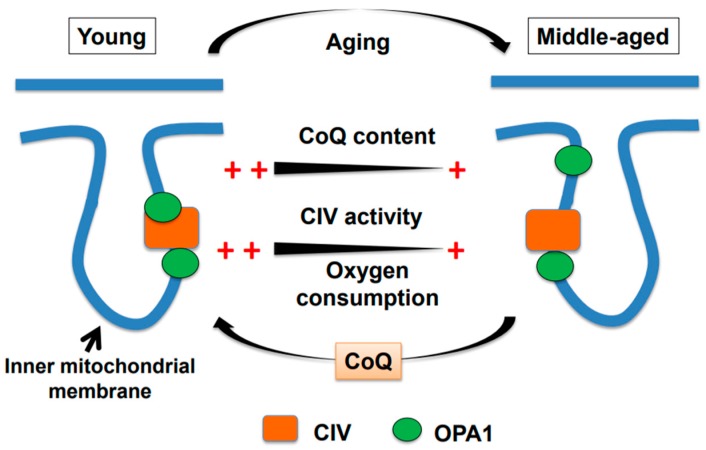
A graphical model of age-associated declines in brain mitochondrial function in male mice. The oxygen consumption rate (OCR) and cytochrome *c* oxidase activity of complex IV (CIV) and coenzyme Q (CoQ) levels are significantly lower in middle-aged mice than in young mice. Concomitantly, the amount of 95-kDa OPA1 bound to CIV is also lower in middle-aged mice. All of these age-associated alterations are restored by the administration of exogenous water-soluble CoQ10 to middle-aged mice.

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
