# Peer review of "Water-soluble CoQ10 as A Promising Anti-aging Agent for Neurological Dysfunction in Brain Mitochondria"

_antioxidants, 2019, doi:10.3390/antiox8030061_

Round 1

Reviewer 1 Report

In this review the authors describe the beneficial effect of exogenous CoQ10 supplementation in neurodegenerative diseases such as PD, Alzheimer and in age related decline of brain functions. the results reviewed in the paper are interesting and the paper can be accepted for pubblication.

minor points: it will be good to indicate for how long the mice were treated with the CoQ10 nanoemulsion. The authors suggest that the beneficial effect of CoQ10 supplementation in middle-aged mice is related to an enhancement of OPA1 binding to mitochondrial Complex IV: can the authors suggest a possible mechanism for the role of CoQ10 in promoting this association?

Author Response

Reply to Reviewer 1.

Minor point 1: Mice were administered 150 mM exogenous CoQ10 for 7 days via drinking water.

Minor point 2: CoQ10 supplementation may cause GTPase-dependent binding of OPA1 to complex IV and activate complex IV, thereby leading to enhancement of the OCR. This assumption is supported by the report describing a loss of OPA1-complex IV interaction in skin fibroblasts from dominant optic atrophy patients with a significant impairment in mitochondrial ATP synthesis (Zanna et al., Brain, 2008). The sentence concerning this has been added in the revised text.

Reviewer 2 Report

This is an interesting paper which discusses CoQ10 as an anti-aging agent.

I have some comments that I would be grateful for the authors to address:

The subject of the review is really the effect of aging in mice on mitochondrial function and I this this should be reflected in the title and then the abstract  should include a sentence to stress that the effect of aging on neurological mitochondrial function in the present review is outlined in mice. 

Since the authors use a specific water solubilised CoQ10 formulation they should explain the rationale for their choice of formulation.

I though some of the text a little confusing (lines 29-31) since the authors state succinate dehydrogenase activity in complex II and cytochrome oxidase activity in complex IV. It would be clearer if the authors wrote ` complex II (succinate:ubiquinone reductase) activity and complex IV (cytochrome oxidase) activity` and once they define the enzymes by their full names continue to use complex IV or II. Complex I should also be define as complex I (NADH: ubiquinone reductase).

I don`t think integrated ETC function can be evaluated by measuring the OCR of complex IV activity as this is only assessing a fraction of the chain. The authors should explain how OCR is measured and change this statement.

In lines 46-47, the OCR of heart is several times higher than other tissues,`  Does this reflect the higher mitochondrial content of heart or more active ETC, not clear from sentence.

In line 75, is OPA1 essential for complex IV activity and does the decline in OPA effect the OCR of the supercomplex ETC enzymes or is it specific to complex IV. Is this discussing neural tissue only or is it a generalised phenomenon?

In section 2.4 is the reduced locomotive ability of middle age mice the result of mito dysfunction induced by a-syn or the effect on neurotransmitter transport? I think it would be good to link and explain a causative mechanism.

In section 2.5, Can you offer a reason for the decline in cerebral CoQ10 with age? Increased oxidative stress etc.

On line 121 it would be good to present the ratio of CoQ9:10 in rats as the majority of the ubiquinone species is CoQ9. Include a sentence of how the ratio of ubiquinol to ubiquione (the reduced active form of CoQ10 and 9) changers with age.

In section 3.1 although exogenous CoQ10 treatment enhances mitochondrial function and increases CoQ10 + 9 status  is this to the levels of young mice and is there any associated evidence of oxidative stress.

In section 3.2 the inc in complex IV activity is suggested to result in enhanced OPA binding, is there any evidence for this or is CoQ10 acting as an antioxidant and protecting inner mito membrane cardiolipin status which is essential for complex IV activity and supercomplex assembly? The authors should discuss this point and provide details about evidence of oxidative stress amelioration following CoQ10 supplementation.

What is the % uptake of water soluble CoQ10 in the brain as it is widely regarded that CoQ10 passes across the blood brain barrier very poorly and this information would be required.

In pages 163-171, does CoQ10 decrease 15d-PG levels to enhance OPA-1 binding? No clear how CoQ10 supplementation enhances OPA1 binding to complex IV. This paragraph requires rewriting to improve clarity.

Throughout the text of the paper the ETC is either written in its full name or denoted as the respiratory chain. It is important that authors define the mitochondrial electron transport chain as ETC at the start of the paper and continue to use the `ETC` throughout the paper.

Author Response

Reply to Reviewer 2:

#1: The title has been changed to “Water-soluble CoQ10 as a promising anti-aging agent for neurological dysfunction in brain mitochondria”.

#2: The reason for our choice of CoQ10 has been added in section 3.1. of the revised text.

#3: According to the suggestion, we have defined the full names of enzymes and complexes in the revised text.

#4: As complex IV is the final enzyme in the ETC and transfers the electrons to oxygen molecules, the most reliable and sensitive methods for evaluating the overall activity of mitochondrial ETC is measuring the OCR in complex IV using high-resolution respirometry. The sentence concerning this has been changed in the revised text.

#5: The reason for high OCR in heart is unknown. Mitochondrial DNA content, which may reflect the mitochondrial content, is highest in heart among the tissues and organs of rainbow trout, including the brain, kidney, liver, red and white muscles, and blood (Leary et al., J. Exp. Biol. 201, 3377, 1998). The reference has been additionally cited.

#6: line 71-75: Age-associated declines in OPA1 binding to complex IV and OCR were the associated events, however, CoQ10-dependent restoration of the OCR required OPA1 binding to complex IV. The reduced OPA1 binding to complex IV was not accompanied by any changes in the supercomplex assembly and whether reduced OPA1 binding to complex IV occurs in a specific type of supercomplex was not determined. A reference reporting the interaction between OPA1 and respiratory complexes in skin fibroblasts from optic atrophy patients has been added.

#7: section 2.4. Alpha-synuclein usually localizes to synaptic vesicles (neurotransmitter vesicles) and plays a role in neurotransmitter release from the synaptic membranes via modulating vesicle pool and vesicle docking with the membrane. On the other hand, VGluT1 is preferentially associated with the membranes of synaptic vesicles and function in glutamate transport. Accumulation of abnormal alpha-synuclein in PD interferes with the priming of synaptic vesicles, which leads to a decrease in the size of the pool of releasable vesicles (Abeliovich and Gitler, 2016). Moreover, the motor cortex is upstream of the corticostriatal circuit or cortex-nigrostriatal circuit connection (Alexander et al., 1986; Kiritani et al., 2012; Shepherd, 2013). Taken together, brain mitochondrial dysfunction that is accompanied by an increase in phosphorylated alpha-synuclein levels together with decreased VGluT1 levels in the motor cortex is suggested to result in motor impairment of middle-aged mice. The sentences concerning this have been added in the revised text.

#8: A reason for age-associated decline in cerebral CoQ remains to be clarified.

#9: The sentences concerning the ratio of CoQ9:10 have been added in the revised text, according to the suggestion.

#10: section 3.1. Exogenous CoQ10 administration to middle-aged mice enhanced the mitochondrial OCR and increased CoQ levels comparable to those in young mice. The sentence concerning this has been added in the revised text. We have not determined whether exogenous CoQ10 induced oxidative stress-related events.

#11: section 3.2. CoQ10-induced OPA1 binding to complex IV, activation of CIV activity, and restoration of OCR were inhibited by 15d-PGJ2. Therefore, it is most likely that exogenous CoQ10 enhances OPA1 binding to complex IV and activates complex IV, thereby leading enhancement of the ETC. Evidence showing the oxidative stress amelioration following CoQ10 supplementation has been provided in the revised text.

#12: Approximately 0.3~0.5% of exogenous CoQ10 was incorporated into the brains of middle-aged mice by the administration of 150 mM water-soluble CoQ10 via drinking water for 7 days. The sentence describing this has been added in section 3.1. of the revised text.

#13: The CoQ10-induced restoration of OCR is “susceptible” to 15d-PGJ2 is unclear, as suggested. Instead, the sentence has been replaced by “The CoQ10-induced restoration of OCR is completely inhibited by 15d-PGJ2”.

#14: According to the suggestion, the “ETC” has been used throughout the paper.

Reviewer 3 Report

Being this a review, the pictures shown in figure 1 should come from an already peer-reviewed paper (please provide permission).

Some sections within the manuscript read a bit dense with disjointed information and would benefit from re-organization of paragraphs.

The functional role of mitochondrial ROS affecting calcium handling proteins involved in contractility (Marks et al. PNAS 2014;111:15250-5) should be briefly discussed.

English language (syntax, grammar, correct choice of words, correct use of adjectives and adverbs) should be substantially improved throughout the text.

Author Response

Reply to Reviewer 3:

#1: Figure 1 has been deleted from the revised text.

#2: Paragraphs have been re-organized in the revised text, according to the suggestion.

#3: In the literature you indicated (PNAS 2014; 111, 15250), the functional role of mitochondrial ROS affecting calcium handling proteins, but not CoQ10. So we have not our position to consider its suitability to discuss in the text.

#4: English language has been edited by “English Language Editing – Springer Nature Author Services”.

Reviewer 4 Report

In this work, Takahashi and Takahashi review the recent findings concerning age-associated declines in mitochondrial function and the positive effects of exogenous water-solubilized CoQ10. This is essentially a review of the authors findings with references to the supporting literature. The manuscript would benefit of a more complete and unbiased review of the available literature, and comparison between different formulation of CoQ10 used and their effects.

Specific comments:

- In the title, “CoQ10 as a promising anti-aging agent” should be changed in “Water-soluble CoQ10 as a…”

-Fig. 1 is not really necessary since readers can not appreciate impaired motor function from a Figure

-Ref 32-34 are wrong; they do not refer to studies in PD patients but to patients with primary CoQ10 deficiency and CoQ10 deficiency secondary to neuromuscular diseases

-References should be added to Sections 3.1, 3.2, and 3.3. Section 3.1 has only one reference (a previous paper from the authors), and Sections 3.2 and 3.3 have no references

-Which are the levels of CoQ9 and CoQ10 in different regions of the brain before and after supplementation, with different formulations of CoQ10, including the water-soluble?

-What are the data available in humans?

Author Response

Reply to Reviewer 4:

In the revised text, we have attempted to cite the available literature and compare between different formulations of CoQ10 as much as possible.

Specific comments:

#1: The title has changed in “Water-soluble CoQ10 as a promising anti-aging agent for neurological dysfunction in brain mitochondria”.

#2: Fig. 1 has been deleted from the revised text.

#3: References concerning PD have been cited, according to the suggestion.

#4: According to the suggestion, references have been added in the revised text as many as possible.

#5: We examined the levels of CoQ in whole brain. Regional difference in CoQ levels was not determined before or after CoQ10 supplementation.

#6: We have investigated on age-associated alterations in brain mitochondrial function and their restoration by exogenous CoQ10 in mice. References describing the data about amelioration of human disorders by water-soluble CoQ10 have been cited in the revised text.

Round 2

Reviewer 4 Report

The authors addressed the issues raised by the reviewers and improved the manuscript